# Development of a Mucoadhesive Vehicle Based on Lyophilized Liposomes for Drug Delivery through the Sublingual Mucosa

**DOI:** 10.3390/pharmaceutics14071497

**Published:** 2022-07-19

**Authors:** María José De Jesús Valle, Aranzazu Zarzuelo Castañeda, Cristina Maderuelo, Alejandro Cencerrado Treviño, Jorge Loureiro, Paula Coutinho, Amparo Sánchez Navarro

**Affiliations:** 1Department of Pharmaceutical Sciences, Faculty of Pharmacy, University of Salamanca, 37007 Salamanca, Spain; mariajosedj@usal.es (M.J.D.J.V.); drury@usal.es (A.Z.C.); cmaderuelo@usal.es (C.M.); alex2@usal.es (A.C.T.); 2Institute of Biopharmaceutical Sciences, University of Salamanca, 37007 Salamanca, Spain; 3CPIRN-IPG—Center of Potential and Innovation of Natural Resources, Polytechnic Institute of Guarda, 6300-559 Guarda, Portugal; jcloureiro97@gmail.com (J.L.); coutinho@ipg.pt (P.C.); 4CICS-UBI—Health Sciences Research Centre, University of Beira Interior, 6200-506 Covilha, Portugal

**Keywords:** liposomes, transmucosal delivery, lyophilized liposomes, liposome permeation, sublingual tablets

## Abstract

A pharmaceutical vehicle based on lyophilized liposomes is proposed for the buccal administration of drugs aimed at systemic delivery through the sublingual mucosa. Liposomes made of egg phosphatidylcholine and cholesterol (7/3 molar ratio) were prepared and lyophilized in the presence of different additive mixtures with mucoadhesive and taste-masking properties. Palatability was assayed on healthy volunteers. The lyophilization cycle was optimized, and the lyophilized product was compressed to obtain round and capsule-shaped tables that were evaluated in healthy volunteers. Tablets were also assayed regarding weight and thickness uniformities, swelling index and liposome release. The results proved that lyophilized liposomes in unidirectional round tablets have palatability, small size, comfortability and buccal retention adequate for sublingual administration. In contact with water fluids, the tablets swelled, and rehydrated liposomes were released at a slower rate than permeation efficiency determined using a biomimetic membrane. Permeability efficiency values of 0.72 ± 0.34 µg/cm^2^/min and 4.18 ± 0.95 µg/cm^2^/min were obtained for the liposomes with and without additives, respectively. Altogether, the results point to the vehicle proposed as a liposomal formulation suitable for systemic drug delivery through the sublingual mucosa.

## 1. Introduction

In recent years, the interest in buccal cavity drug administration has progressively increased, and this is nowadays considered an alternative to oral drug administration for population groups with swallowing difficulties, such as the elderly or children, as well as patients with nausea [1,2]. Better acceptability and compliance, the lack of hepatic first-pass metabolism, or easier removal of the formulation, if necessary, are among the advantages of this alternative route compared to conventional peroral administration [3,4]. For systemic effects, drugs must reach the capillary network underlying the mucosa, which is fundamentally achieved by crossing the non-keratinized epithelium located in the buccal or sublingual regions. Because of its optimal blood supply, the sublingual region has been associated with drugs requiring a rapid onset of action, while other regions have been associated with sustained drug delivery [5]. This conception has remained predominant during the development of the different dosage forms currently available on the market for drug administration in the oral cavity [6,7,8]. In this regard, the development of sustained drug delivery systems aimed at the sublingual route faces certain drawbacks that may compromise an effective buccal absorption, such as involuntary swallowing of the formulation or high salivary turnover. The residence time of the formulation in the absorption site is a crucial issue for buccal formulations aimed to produce systemic effects. One of the most attractive strategies for ensuring a proper and long-lasting retention of the formulation on the oral mucosa is based on the use of mucoadhesive polymers, capable of establishing molecular interactions with mucosa components, allowing a durable anchorage of the formulation [9,10,11]. Polymeric and lipid nanoparticles are suitable nanocarriers for buccal drug administration [12,13], and liposomes stand out for their high biocompatibility and versatility, allowing these lipid vesicles to be loaded with hydrophilic or hydrophobic drugs. Moreover, liposomes can come across body membranes and transport drugs to deeper structures near the capillary network [14,15]. The lipophilic nature of liposomes enables them to penetrate membranes via a transcellular route with four different mechanisms, including adsorption, lipid exchange, fusion and endocytosis [16,17].

Buccal drug delivery systems based on liposomes have been developed for some products with good results. Vitamins or drugs showing a relevant first-pass effect due to high hepatic metabolism, even immunogenic proteins, have been proposed for buccal administration [18,19,20,21], and the possibility of using liposomes for insulin administration through the oral mucosa has been reported recently [22,23]. However, liposome instability hinders drug formulation development based on this type of lipid vesicle, and lyophilization is recommended for long-term stabilization [24,25]. Lyophilization presents a great potential in the pharmaceutical field not only as a technique for the preservation of thermolabile drugs, but also as a technological process for developing drug delivery systems based on nanoparticles. This technique produces oral disintegrating tablets and has been recently applied to produce a mucoadhesive matrix and wafers for buccal drug delivery [26,27,28]. The advantages of the sublingual cavity for systemic drug absorption combined with the benefits of liposomes as drug carriers are a promising opportunity for developing innovative pharmaceutical formulations with application to new and old drugs. Therapeutic agents showing inefficient and erratic oral absorption would benefit from the synergistic effects of the above combination as far as formulation anchorage and liposome stabilization are achieved. We hypothesize that liposomes can be stabilized by lyophilization and included in mucoadhesive formulations that enable rehydration and release to permeate through the buccal mucosa and reach the inner tissue layers close to the capillary network. According to the above considerations, the main objective of this work was to develop a mucoadhesive vehicle based on lyophilized liposomes for systemic drug delivery through sublingual mucosa. The proposed vehicle might be applied to drugs with inefficient and erratic oral bioavailability, irrespectively of the drug being a new therapeutic agent or an approved product already used in clinical practice. Even for drugs with high oral bioavailability, this vehicle might provide a beneficial alternative for patients with swallowing difficulties needing pharmacological treatment.

## 2. Materials and Methods

### 2.1. Materials

L-α-phosphatidylcholine egg yolk (EPC) and cholesterol (Ch) were purchased from Sigma-Aldrich^®^ (Merk KGaA, Darmstadt, Germany) laboratory. HPLC-grade methanol was supplied by Thermo Fisher Scientific. H_2_PO_4_, NaOH, disodium phosphate and citric acid monohydrate were purchased from PanReac ApplieChem (Darmstadt, Germany). Ultrapure water was obtained with a Wasserlab Automatic Plus System. Lactose monohydrate (L), mannitol (M), sodium alginate (A) and carboxymethylcellulose sodium 1500–4500 (C) were purchased from Guinama S.L.U. (Valencia, Spain).

### 2.2. Methods

#### 2.2.1. Liposomes

Transmembrane pH gradient liposomes were prepared following a previously described method [29] with some changes. In brief, EPC and Ch (molar ratio 7/3) were mixed with pH = 4.7 citrate buffer solution to a total lipid concentration of 1.8% *w*/*v*. The mixtures were placed in a Fisher Scientific FB 15061 ultrasonic bath (50 Hz) at 40 ± 2 °C for 20 min. The resulting suspensions were filtered 8 times across syringe filters (Cromafil^®^ PET) of 0.45 µm pore size, and the filtered samples were kept at room temperature for 60 min and then stored at 4 °C for 1 h. NaOH 1 N was added to the liposome samples until pH = 7.0 ± 0.1 and maintained for 20 h at 4 °C in a shaking water bath (Unitronic OR Selecta P) to produce the transmembrane pH gradient. Under ultrasonic agitation at 40 °C (above transition temperature), EPC molecules are in the liquid state with mobility for bilayer arrangement and vesicle formation. The 7/3 ratio was selected based on previous studies carried out in our laboratory (unpublished data) that are in accordance with results from Mare et al. confirming the benefits of this bilayer composition [30].

#### 2.2.2. Mixtures

Lactose alone (L) or combined with mannitol 1:1 *w*/*w* ratio (M + L) at a total concentration of 4% (*w*/*w*) were used as taste-masking additives and lyoprotective agents (additive/EPC mass ratio = 2.6). Sodium alginate alone (A) or combined with carboxymethylcellulose 1:1 *w*/*w* ratio (A + C) were used as gelling-mucoadhesive agents at a total concentration of 0.4% (*w*/*w*). First, L or L + M were added to liposome suspension, and mixtures were stirred for 5 min. Then, the mucoadhesive agent (A or A + C) was added to the mixtures, and the resulting samples were kept under slow agitation during the gelling process. Figure 1 shows the combination of additives in mixtures.

Samples were distributed in blister (2 g) or glass vials (2 g) and lyophilized using Telstar Cryodos laboratory freeze-drying equipment. Freezing at −80 °C and primary drying at −40 °C for 72 h with condenser temperature −80 ± 4 °C and chamber pressure 0.008 ± 0.002 mBar were applied to produce preliminary cakes for testing palatability and selecting additives.

#### 2.2.3. Size, Polydispersity Index and Zeta Potential

Hydrodynamic diameter (Dh), polydispersity index (PDI) and zeta potential of fresh liposomes with and without additives were analyzed by dynamic light scattering (DLS) using a Zetasizer Nano ZS (Malvern Instruments, Co., Malvern, UK). The analysis was performed at 25 °C and a scattering angle of 173° after the appropriate dilution with Milli-Q water to avoid multiple scattering. The same analysis was performed with fresh and rehydrated samples.

#### 2.2.4. Scanning Electron Microscopy

The morphology of liposomes before and after lyophilization and liposomes released from tablets during in vitro studies was analyzed by scanning electron-transmission microscopy (SEM) using a JSM-IT500 InTouchScope™. Samples were fixed with poly-L-lysine and osmium, and acetone was used as the desiccant.

#### 2.2.5. Viscosity

The dynamic viscosity (η) of liposome suspension with and without additives was determined at 25 ± 2° C. A rotary viscosimeter with the rotor and rotation rate selected according to the sample’s resistance was used, and the viscosity was estimated from the equation:η = (N/A)/γ
where N/A is the force per unit area, and γ is the shear rate.

#### 2.2.6. Palatability Assay in Healthy Volunteers

An in vivo palatability assay was conducted with the preliminary freeze-dried products by a randomized, incomplete, crossover, balanced, single-blind design. For this purpose, 15 healthy adult volunteers aged 23–65 years (80% female) were recruited, and informed consent was obtained in accordance with the Helsinki declaration. Each participant received two of the six formulations tested and a questionnaire with the items to be assessed: smell, flavor, texture and buccal sensation. Essentially, after rinsing the mouth with water, the formulation was placed on the tongue until no product was left. After assessing the first formulation, participants were asked to rinse their mouths twice with water (palate cleanser) and proceed the same way for the next formulation. A visual analogue scale (VAS) was used to rate each item (Figure 2).

#### 2.2.7. Differential Scanning Calorimetry (DSC)

The glass transition temperature (Tg) and glass transition temperature of maximum freezing concentration (T′g) of the mixtures, as well as the phase transition™ of lyophilized samples, were determined by DSC. Experiments were performed using a Mettler Toledo DSC-1. Samples were weighed into 40 µm aluminum pans that were hermetically sealed, and an empty sealed pan was used as a reference. Once the pans were placed on the sample chamber, the following program was run to determine Tg and T′g: cooling from room temperature to −35 °C, maintenance of −35 °C for 10 min and heating to 25 °C. Ramping was at 10 °C/min for both cooling and heating. To determine Tm, lyophilized samples were scanned from 25 °C to 150 °C at 10 °C/min. The analysis of the thermograms was performed with the STARe software package. Tg, T′g and Tm were considered the onset of the corresponding peak during the cooling or heating scan.

#### 2.2.8. Optimization of the Freeze-Drying Process

The mixture selected from the results of the palatability assay (Lip + L + M + A) was subjected to a series of experiments to determine the optimum conditions for lyophilization. All experiments with this mixture were carried out using an LYOBETA 6 PL (Tesltar) with process qualification certificate, connected to a programmable logic controller (PCL) and MicroSuiteLab control software. The freeze dryer had four shelves with a total usable shelf area of 0.9 m^2^ and a Pirani gauge for chamber pressure (Pc) control. Product temperature measurement was performed every 15 min. Samples were placed in the trays, avoiding the front edge to reduce the effect of the atypical radiation due to the acrylic door.

T′g value obtained by DSC analysis for Lip + L + M + A was considered for selecting the target product temperature during primary drying (*target* T_p_). The ramp temperature approach previously described by Assegehegn et al. [31] was applied to select the fluid temperature (T_f_) and the duration of primary drying. The first experiment was performed under the following conditions: freezing temperature at −60 °C, primary drying at −30 °C for 24 h and secondary drying for 12 h at 10 °C followed by 6 h at 18 °C. Another run was performed under the following conditions: freezing temperature at −45 °C, staggered temperatures (−25 °C, −22 °C, −20 °C and −17 °C) for 4 h each for primary drying and secondary drying for 12 h at 15 °C. A third run was performed under the following conditions: freezing temperature at −40 °C, primary drying at −17 °C for 24 h preceded by 2 h at −20 °C and secondary drying for 18 h at 17 °C. A pressure of 100 µBar was selected for the first period of primary drying (4 h) and 25 µBar for the rest of the process. The ramp temperature was 1 °C/min in all cases, irrespective of primary or secondary drying (recipes of cycles in Appendix A). The relationship between the fluid and product temperature (T_f_/T_p_) during primary drying was used to select the best conditions for this phase. With respect to the secondary drying, the remaining moisture (RM) in the final products was analyzed and used for selecting optima conditions. The RM in lyophilized cakes was determined by the Karl Fisher method, transferring 0.1 g of sample (Mettler Toledo XS 105DU) to the titration cell. The volumetric water content was measured using a Metrohm 870 KF Titrino plus KF titrator, and the results are shown as RM *w*/*w* value on the dry product bases. The optima conditions were then applied to produce the lyophilized product for tablet manufacturing.

#### 2.2.9. Mucoadhesive Tablets

Mucoadhesive tablets were prepared on an eccentric tablet press (Bonals BMT, Bonals, Spain) equipped with two punch geometries: (a) round, 11 mm and (b) capsule-shaped, 22.4 mm × 6.2 mm. Tablets were produced by compression of two blister cakes, one immediately after the other. Changes in the position of the inferior and superior punch allowed for different compression forces that were tested until finding the proper one, producing compact tablets suitable for buccal administration. The resulting tablets were assumed to be bidirectional, and unidirectional tablets were prepared by the additional compression of 0.04 g of alginate powder on one side of the bidirectional tablets.

##### Weight and Thickness Uniformity

Twenty and ten tablets were randomly selected and individually tested for weight and thickness, respectively, using a Mettler Toledo XS 105 balance and an Erweka TBH 210TD durometer, respectively. Results are expressed as the average ± standard deviation (SD) in both cases.

##### Swelling Assay

An in vitro study was conducted to test unidirectional and bidirectional tablets’ swelling index (SI). Simulated salivary fluid (SSF), composed of mineral salts, Tween^®^ 20 and xanthan gum [32], was used for the assay. Each tablet was weighed and put into a beaker containing 5 mL of SSF. Beakers were placed in an incubator at 36 °C and 60 rpm shaking rate. Tablets were re-weighed at predetermined time intervals (0.25, 0.5, 1, 2, 3, 4 and 6 h), and the swelling index (SI) was calculated by using the following equation:SI (%) = [(Wt − Wi)/Wi] × 100
where Wt is the tablet’s weight at each interval time, and Wi is the initial weight of the tablet.

##### Tablet Test in Healthy Volunteers

An in vivo assay in healthy volunteers was conducted to evaluate round and capsule-shaped tablets to select the best performing geometry. Six participants, 23–65 years (84% female), were recruited for a randomized, complete, crossover, balanced, single-blind design assay, and their declarations of informed consent were obtained in accordance with the Helsinki declaration. Each participant received two tablets and a questionnaire with the items to be assessed: adhesion time, adhesion strength and signs of irritation. Briefly, they had to rinse their mouth with water and then place the tablet in the sublingual area of the oral cavity, pressing until attachment. During the test, they were asked to refrain from drinking or eating, but they were allowed to undergo regular activity. A 24 h washout period was fulfilled before testing the remaining formulation.

##### In Vitro Liposome Release

Tablets of the selected geometry (round tablets) were subjected to in vitro assays to evaluate liposome release. A first assay was performed using a round-bottom glass tube containing 5 mL of SSF at 36 °C. The tablets were placed at the bottom of the tube and lightly pressed for adhesion. In the unidirectional tables, the alginate-coated side was placed up to evaluate its barrier effect. Glass tubes underwent agitation for 45 min, and SSF samples were taken for cholesterol quantification using the HPLC technique described below. Another in vitro assay was carried out using the USP apparatus. Tablets were stuck to the bottom of the vase containing 300 mL of PBS pH 7 at 36 °C with paddles at 100 rpm. Unidirectional tables were placed with the alginate-coated side up, as in the previous in vitro assay. PBS samples were taken at the previously programmed times (0.5, 1, 2, 3, 4, 6, 8 and 12 h) for cholesterol quantification. The unidirectional and bidirectional tablet curves were compared by estimating the similarity factor (f2) [33].

##### Permeation Assay

Permeation efficacy of liposomes was studied using vertical diffusion chambers with a permeation area of 2.54 cm^2^. The sample (liposome suspension with or without additives) was placed in the donor compartment, and a PermeaPad^®^ membrane of 25 mm diameter was mounted to reach occlusive and infinite dose conditions. Following this, the chamber was immersed in a vase containing 100 mL of PBS at 36 °C with a 250 rpm paddle agitation as the receptor compartment. PBS samples were taken from the receptor compartment at 2 min intervals during the first 10 min and at 10 min intervals for the rest of the experiment. The cholesterol in the withdrawn samples was quantified by an HPLC technique described below, and the accumulated amount of cholesterol in the receptor compartment was estimated. The cumulative amount of permeated cholesterol (Qt) was plotted against time, and the slope of the linear part of the curve, representing the steady-state flux rate (Jss), was used for permeability (P) calculation, according to the following equation:P (µg/cm^2^/min) = Jss/A
where Jss is the slope of the linear fraction of the curve, and A is the permeation area.

##### HPLC Technique for Cholesterol Quantification

Chromatographic analysis was performed using a previously described method [34]. In brief, a Hypersil BDS C18 (25 cm × 4.6 mm i.d., 5 µm particle size, 80 A pore size) column was used. Furthermore, a 100% HPLC-grade methanol isocratic mobile phase at a 1.0 mL/min flow rate and run time of 10 min was applied. Column and sample temperatures were 40 °C and 20 °C, respectively. The diode array detector was operated at 210 nm with 4 nm bandwidth. The injection volume was set at 50 µL, and the samples were diluted with methanol containing 0.5% *v*/*v* formic acid before HPLC injection. The calibration range tested was 1 to 50 μg/mL. The coefficient of determination was above 0.999, the deviation from the true value (mean accuracy) and the relative standard deviation (precision) were below 10%, while the limit of quantification was below 0.5 μg/mL (signal/noise 12.24 and CV < 10%).

### 2.3. Statistical Analysis

A comparison of results registered as mean values was performed using a one-way analysis of variance (ANOVA) with Tukey´s post-hoc test. Statistical significance was considered for *p*-values ≤ 0.05. For kinetic profile comparison, the similarity factor was estimated, and values under 50 were considered statistically significant.

## 3. Results and Discussion

The eco-friendly and solvent-free method applied here produced liposomes of Dh = 265.83 ± 12.05 nm, PDI = 0.27 ± 0.01 and zeta potential −46.77 ± 1.61 mV. After mixing with additives, Dh and zeta potential did not significantly change, and PDI slightly increased, although differences were not statistically significant (*p* = 0.1444) (Table 1). Highly negative zeta potential values were observed in this study, despite PCs being neutral lipids. Nevertheless, extensive literature data confirm the negative surface of PC liposomes prepared from commercial products, which are not pure PC but a mixture of lipids. The product used here had ~70% of PC, and this may be the reason for the highly negative zeta potential found.

Concerning viscosity, samples with alginate showed higher values than carboxymethylcellulose, irrespective of containing L (51.75 ± 11.67 cP vs. 21.45 ± 5.03 cP) or L + M combination (53.00 ± 24.04 cP vs. 29.30 ± 1.11 cP).

Taste is one of the primary determinants of market performance and commercial success of buccal formulations and is crucial for patient compliance [35]; therefore, the masking ability of sweeteners or flavoring agents is highly recommended to be scrutinized during early formulation development. Since EPC has an unpleasant smell and taste, the mixtures were lyophilized under preliminary conditions to obtain cakes suitable for testing palatability in healthy volunteers. Lactose was used in this study because previous work had proven its lyoprotective effects for liposomes with compositions similar to those prepared here [29]. The benefits of mannitol as a bulking agent and suiter product are well documented [36,37,38]. The results of the palatability assay (summarized in Appendix A) show that all mixtures were positively evaluated. However, the combination of lactose and mannitol produced a taste-masking effect superior to that observed for each single product. Figure 3 illustrates the results of the best-evaluated mixture, which was the one containing lactose, mannitol and alginate.

According to the palatability assay, Lip + M + L + A was the optimal composition; therefore, it was used for the rest of the experiments.

The results from the DSC studies are shown in Figure 4. Thermal analysis revealed that Tg was dependent on the additives and the Lip + A mixture showed the lowest Tg value (−20 °C), while the Lip + L + M + A mixture showed the highest value (−15.83 °C). For T′g, however, differences between liposomes with and without additives were not found (−5.95 to −5.03 °C range).

The DSC results showed a T′g value of –5.95 °C for the Lip + L + M + A mixture, and the collapse temperature (Tc) assumed to be 2–3 °C above the T′g [39] was −4 °C. Accordingly, up to −4 °C might be safely reached during primary drying for our product. Nevertheless, −8 °C was selected as the target T_p_ for primary drying to ensure the avoidance of product collapse. T_f_ and T_p_ ramps for three cycles are shown in Figure 5 and were analyzed to select the optimal conditions to produce the lyophilized material used for the mucoadhesive tablets.

T_f_ and T_p_ ramps revealed the magnitude of the difference between both temperatures all along the cycle. T_f_ = −40 °C and −17 °C were selected for the freezing step and primary drying, respectively—the former to make sure that the product was cooled under its Tg (−15.83 °C) and the latter to ensure that T_p_ did not reach the target (−8 °C) during primary drying. With respect to the secondary drying, it was found that the increased temperature from 17 °C to 20 °C did not produce significant differences in the RM of lyophilized samples. However, values above 3% were obtained in all cases. A fluid temperature of 17 °C was finally selected for secondary drying. Accordingly, the following conditions were selected for lyophilization: freezing at −40 °C for 4 h, temperature ramp to −20 °C at 1 °C/min, maintenance at this temperature for 1 h, temperature raised to −17 °C and primary drying at −17 °C and 25 µBar for 24 h. Then, the temperature was ramped to 17 °C at 1 °C/min and secondary drying at 17 °C and 25 µBar for 18 h. This cycle was applied to obtain the lyophilized cakes used for preparing the mucoadhesive tablets. Uniform and elegant cakes were obtained under these conditions, and these cakes were easily removed from the blisters without crumbling or deforming, as Figure 6 shows. RM values of 3.1 ± 0.4% *w*/*w* were found in the lyophilized cakes.

It is worth noting that samples showed a remarkable behavior characterized by a high amount of unfrozen water remaining in the lyophilized cakes after long-lasting secondary drying periods. It has been reported that physical bonds between alginate chains and water molecules are responsible for non-freezing water content in alginate aqueous mixtures, which is highly dependent on sample composition [40]. Samples without liposomes but containing just mannitol, lactose and alginate also showed high RM (5.2 ± 0.2% *w*/*w*), confirming the gelling agent contribution to the non-freezing and strongly bound water found in lyophilized samples.

A Tm value of 52.16 ± 0.05 °C was found for the samples lyophilized under the above conditions (Figure 7).

Tm is related to the stability of lyophilized products. After freeze-drying, amorphous components remain vitrified only if the storage temperature is well below the transition temperature. As a rule of thumb, a difference of 45–50 °C between storage temperature and Tm has been proposed for sample preservation [41]. According to this rule, our lyophilized samples should not be stored at room temperature, but refrigeration under 10 °C would be required for appropriate preservation. These results are in accordance with previous data [29], showing that liposome samples containing lactose showed transition temperatures high enough to be stored at room temperature and that the incorporation of mannitol produced a decrease in the transition temperature. However, palatability was the selection criteria in this study, aiming to develop a pharmaceutical vehicle for sublingual administration.

Round and capsule-shaped tablets were obtained from lyophilized cakes, and all were assayed in terms of weight, thickness uniformity and swelling index. Results in Table 2 show a high uniformity for weight and thickness, irrespective of the tablet geometry. Regarding swelling, tablet weight increased progressively until disintegration (shown in Appendix A) and SI values of 116.72 ± 70.29% and 105.84 ± 15.32% were obtained for capsule-shaped and round bidirectional tablets, respectively.

Differences in table surface areas (138.88 mm^2^ and 95.03 mm^2^ for capsule-shaped and round, respectively) did not produce significant differences in SI (*p* = 0.9881) or the swelling profiles shown in Figure 8, with an f2 value of 99.

The geometry’s influence on the tablet’s accommodation and permanence in the sublingual cavity was evaluated in healthy volunteers who reported that round tablets were more comfortable, easier to adhere to the buccal epithelium and retained longer in the sublingual cavity (results summarized in Appendix A). Taken together, the data point to the round tablets being superior to the capsule-shaped ones. Therefore, the round tablets were selected to perform the rest of the liposome release and permeation studies. Since the mucoadhesive vehicle proposed here is aimed at systemic delivery, and unidirectional tablets are beneficial for this objective, unidirectional tablets were prepared from bidirectional ones by adding an alginate layer on one of the tablet sides. Direct compression of alginate powder produced better results than compression of the same amount as a lyophilized cake, the latter leading to tablets showing a tendency to film detachment.

As reported for bidirectional tables, unidirectional ones showed high uniformity for weight and thickness with mean values of 0.34 ± 5.53 × 10^−3^ g and 3.89 ± 1.03 × 10^−1^ mm, respectively. These values (in Table 2) are slightly higher than observed for bidirectional tables, showing increases of 13.33% and 3.18% for weight and thickness, respectively. With respect to swelling, SI = 221.43 ± 7.11 was found. Significant differences between unidirectional and bidirectional tables were found for SI (*p* = 0.0037), but not for swelling profile (f2 = 86). Figure 9 shows the swelling profiles for unidirectional and bidirectional tablets.

Regarding liposome release, in vitro assays showed that rehydrated liposomes are released from the tablets and that the alginate layer prevented the liposomes from moving across this layer. Comparison of unidirectional and bidirectional tablets showed significant differences in the amount of cholesterol delivered to the media after 45 min of vigorous agitation. For the unidirectional tablets with the alginate layer orientated to the top, cholesterol was not accurately quantified in the media (limit of quantification of the HPLC analytical technique 0.5 µg/mL), while a cholesterol concentration of 22.09 ± 3.07 µg/mL was determined for tablets without the alginate layer. On the other hand, release profiles obtained from the experiments using the USP apparatus (Figure 10) confirmed the barrier effect of the alginate layer. Values under 0.85 ± 0.04 µg/mL were detected all along the experiment for unidirectional tablets, while progressively higher values were observed for bidirectional tablets to reach a maximum of 4.40 ± 0.84 µg/mL.

Cholesterol concentrations in PBS were assumed to be representative of the cumulative liposome released to the media. SEM images of SSF and PBS samples (Figure 11) showed the presence of nanoparticles with the same morphology as found for original liposomes. This finding supports the assumption that cholesterol quantified in PBS is a subrogate marker of the liposome released to the media.

This assumption applies only to early samples due to cholesterol autoxidation in aqueous media. This explains the profile of the cumulative curves showing a descending phase after 4 h. Non-enzymatic oxidation of cholesterol in different sample types was reported [42], and this is one reason that justifies the interest in liposomal formulations being lyophilized for long-term stabilization.

The liposome potential for permeation through the buccal epithelium was evaluated by an in vitro assay. Figure 12 shows the linear portion used for Jss determination. Liposome suspension without additives was assayed and compared to liposomes in the Lip + L + M + A mixture. PermeaPad^®^ membranes were used in this study because the myelin-like structures formed by phospholipids between cellulose membranes in contact with water mimic the buccal epithelium [43]. These membranes have proven functionality in predicting buccal epithelium transport [44]. On top of that, an excellent correlation between PermeaPad^®^ and sublingual mucosa has been found for buccal formulations [45].

The regression analysis of permeation profiles showed that the permeability efficiency for the liposome suspension without additives was much higher than for liposomes in the mixture (4.18 ± 0.95 µg/cm^2^/min and 0.72 ± 0.34 µg/cm^2^/min, respectively), the differences showing statistical significance (*p* = 0.0071). These results reveal that the liposomes could rapidly equilibrate through the membrane and reach the linear phase of the cumulative curve at the receptor compartment. On the other hand, data showed that the additive mixture was able to retain the liposomes and slow their diffusion, with the final result of a permeation rate for the liposomes in the mixture slower than that observed for liposomes in aqueous suspension. The higher viscosity of the mixture (53.00 ± 24.04 cP vs. <3 cP) undoubtedly contributed to the slower diffusion. Furthermore, the gel matrix entrapping the liposomes likely contributed to the permeation profile controlled by the liposome release rate. The finding of liposomes with a high permeation efficacy is relevant since this has not been previously reported, and most studies focus on the permeability of drugs. Nevertheless, one advantage of liposomes is their ability to penetrate membranes and transport-loaded drugs into tissues. The results of this study confirmed that the liposomes prepared here could permeate a membrane that mimics the buccal epithelium and that permeation was not the limiting step. However, the liposome release rate was the step limiting the transport of liposomes through the buccal epithelium.

## 4. Conclusions

The above results prove that the biocompatible and biodegradable tablets produced here have characteristics (palatability, size, placement, comfortability and retention) adequate for sublingual administration. Moreover, it has been proved that the tablets in contact with aqueous fluids swelled, allowing the lyophilized liposomes to be rehydrated and released to the surrounding media. Furthermore, the liposomes showed a high permeability efficiency, making it possible for the transepithelial transport to be limited by the formulation composition controlling the liposome release. For the long-term stability of these liposomes, the results advise storing the formulation under refrigerated conditions since the Tm values are under 60 °C. Studies with prototype drugs loaded into the liposomes are the next step for the further development and clinical application of the pharmaceutical vehicle proposed.

## Figures and Tables

**Figure 1 pharmaceutics-14-01497-f001:**
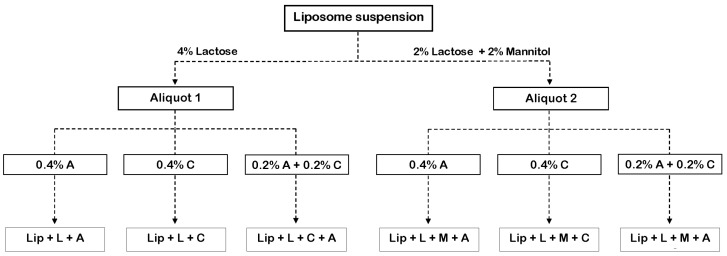
Liposome mixtures assayed for palatability. (A: sodium alginate; C: carboxymethylcellulose; L: lactose; M: mannitol).

**Figure 2 pharmaceutics-14-01497-f002:**
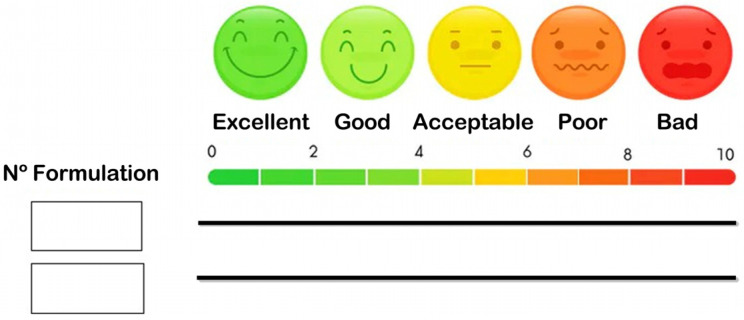
Visual analogue scale used for the in vivo assay of palatability.

**Figure 3 pharmaceutics-14-01497-f003:**
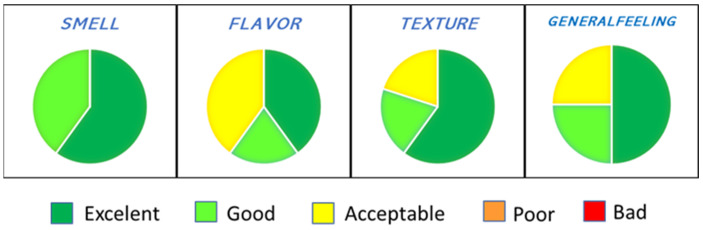
In vivo palatability results obtained for Lip + M + L + A, which resulted as the best-evaluated mixture.

**Figure 4 pharmaceutics-14-01497-f004:**
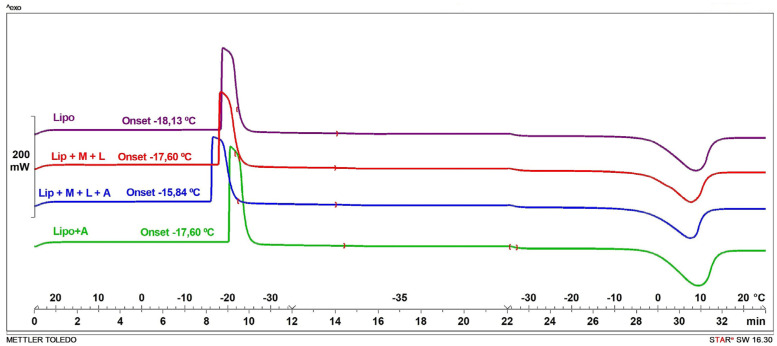
Thermograms of liposome samples without additives and liposomes in mixtures containing different additives (L: lactose; M: mannitol; A: alginate).

**Figure 5 pharmaceutics-14-01497-f005:**
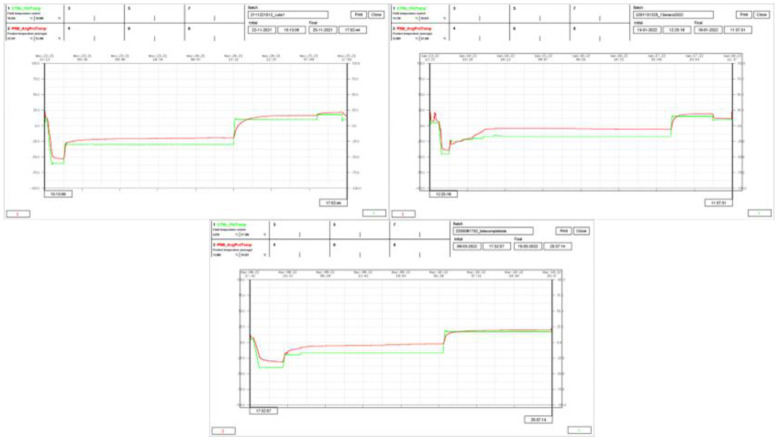
Temperature ramps corresponding to three lyophilization cycles. Green for fluid temperature and red for product temperature.

**Figure 6 pharmaceutics-14-01497-f006:**
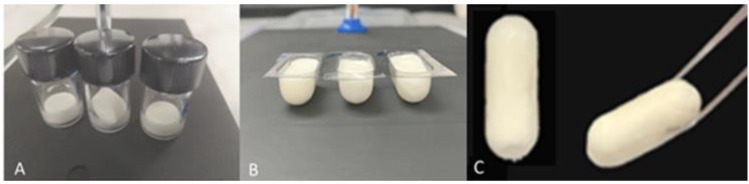
Aspect of lyophilized cakes inside vials (**A**), inside blisters (**B**) and after withdrawal from blister (**C**).

**Figure 7 pharmaceutics-14-01497-f007:**
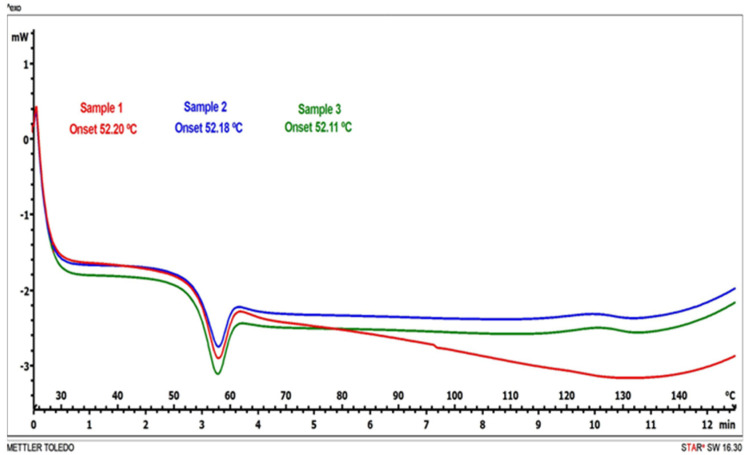
Thermograms obtained by DSC of lyophilized samples (3 replicates).

**Figure 8 pharmaceutics-14-01497-f008:**
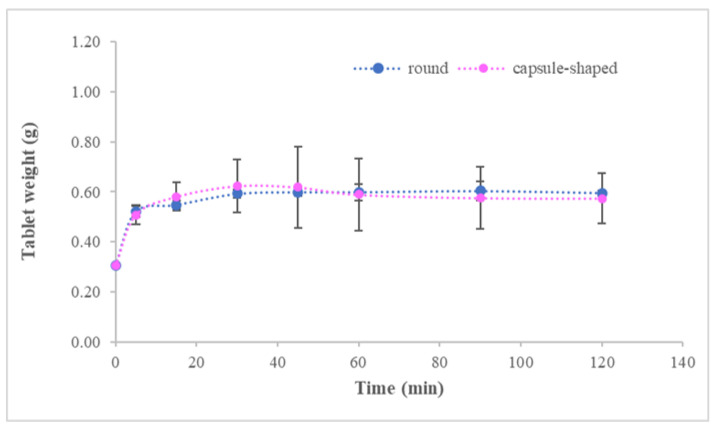
Mean swelling profile obtained for capsule-shaped and round bidirectional tablets. *n* = 3.

**Figure 9 pharmaceutics-14-01497-f009:**
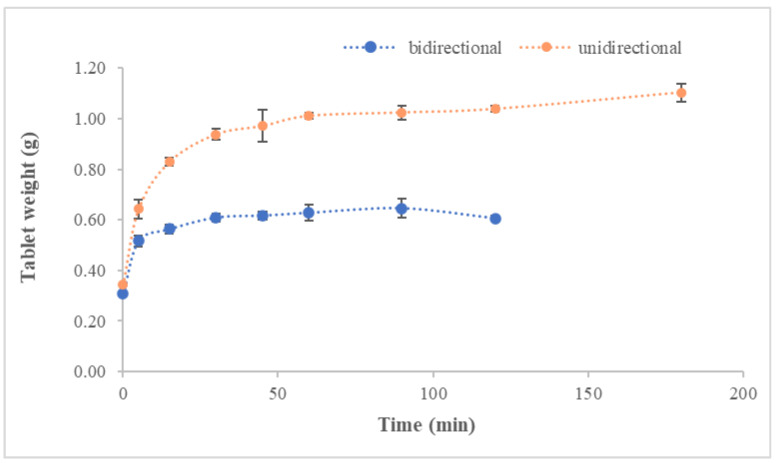
Swelling profiles obtained from unidirectional and bidirectional round tablets. *n* = 3.

**Figure 10 pharmaceutics-14-01497-f010:**
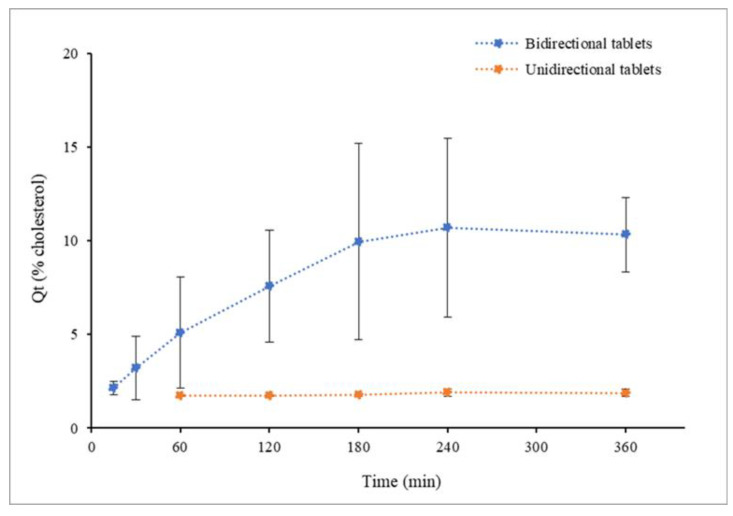
Liposome release from tablets measured as mean curves of cumulative percentage of cholesterol. *n* = 3.

**Figure 11 pharmaceutics-14-01497-f011:**
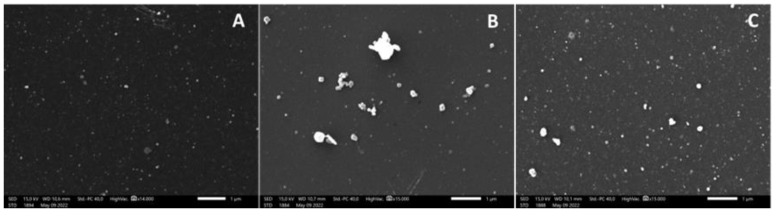
SEM images of original and rehydrated liposomes in SSF and PBS from in vitro assays ((**A**) original liposomes; (**B**) SSF from swelling assay; (**C**) PBS from release assay).

**Figure 12 pharmaceutics-14-01497-f012:**
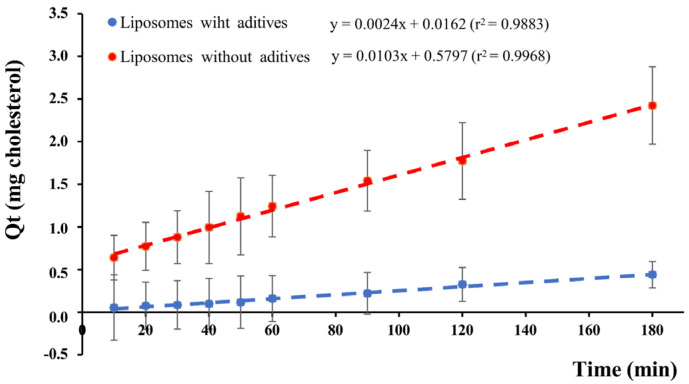
Linear portion of the profiles used for the steady-state flux (Jss) determination (mean ± SD). *n* = 3.

**Table 1 pharmaceutics-14-01497-t001:** Characteristic of liposomes without additives (Lip) and liposomes in mixtures containing different additives (L: lactose; M: mannitol; C: carboxymethylcellulose; A: alginate). Three replicates.

	Viscosity (cP)	Dh (nm)	PDI	Zeta Potential (mV)
Lip	<3	265.83 ± 12.05	0.27 ± 0.01	−46.77 ± 1.61
Lip + L + A	51.75 ± 11.67	286.93 ± 32.71	0.31 ± 0.03	−49.80 ± 1.04
Lip + L + C	21.45 ± 5.03	257.80 ± 9.09	0.28 ± 0.01	−40.50 ± 6.03
Lip + L + A + C	45.00 ± 21.21	274.23 ± 10.71	0.31 ± 0.03	−40.37 ± 3.50
Lip + L + M + A	53.00 ± 24.04	265.50 ± 7.11	0.32 ± 0.03	−44.33 ± 6.82
Lip + L + M + C	29.30 ± 1.11	274.20 ± 18.18	0.33 ± 0.08	−45.17 ± 1.20
Lip + L + M + A + C	46.50 ± 6.20	280.43 ± 18.35	0.37 ± 0.06	−46.93 ± 4.92

**Table 2 pharmaceutics-14-01497-t002:** Characteristics of capsule-shaped and round tables produced from lyophilized cakes. *n* = 20 weight, *n* = 10 thickness and *n* = 3 swelling index and time.

	Bidirectional 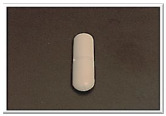 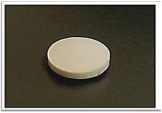		Unidirectional 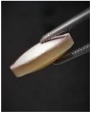
	Capsule-Shaped	Round	Round
**Weight (g)**	0.30 ± 5.81 × 10^−3^	0.30± 7.53 × 10^−3^	0.34 ± 5.53 × 10^−3^
**Thickness (mm)**	3.34 ± 3.35 × 10^−1^	3.77 ± 6.14 × 10^−2^	3.89 ± 1.03 × 10^−1^
**SI (%)**	50.88 ± 13.82	51.23 ± 3.69	68.88 ± 0.69
**Swelling time (min)**	40–60	40–60	180

## Data Availability

The data presented in this study are available within the article.

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
