# Peer review of "Development of a Mucoadhesive Vehicle Based on Lyophilized Liposomes for Drug Delivery through the Sublingual Mucosa"

_pharmaceutics, 2022, doi:10.3390/pharmaceutics14071497_

Round 1

Reviewer 1 Report

The submitted manuscript deals with development and evaluation of tablets containing liophylized liposomes for sublingual drug delivery. The topic is interesting and fits under the scope of the Journal. However, there are some deficiencies that should be adequately addressed to strengthen the paper.

Line 16: Check the amount of egg phosphatidylcholine and correct.

Some numerical data should be added in the abstract to confirm the results obtained.

Liposomes cannot be produced only by dispersing phosphatidylcoline and cholesterol in water. This part in the preparation method is not clear and need clarification. How could you be sure that the liposomes are formed? Is there any picture showing the formed liposomes? Moreover, provide more data about the content of phosphatidylcholine in egg phosphatidylcholine obtained from Sigma Aldrich.

Figure 1: to increase the clarity of presentation, please explain meaning of abbreviations used in the figure in the figure’s caption (legend).

The DLS and ZP measurements were done in water, why not in medium simulating sublingual application?

Line 250: indicate the pH of PBS

The abbreviation for size (line 124) is Dh, later in Results and discussion (line 289) abbreviation Dm corresponds to size. Please correct. Does the term size correspond to average (mean) diameter? If so, then mean diameter instead of size should be denoted in subchapter 2.2.3.

Tables: indicate n value for results expressed as average ± S.D.

Have you checked the size of liposomes formed after disintegration of tablets? Liposomes are formed in situ after disintegration of the tablets, and then, the effect of liposomes could be expected. Do you expect disintegration of tablets? The authors performed in vitro release studies.

What is the goal of the developed system, slow or rapid release, what do you expect from the liposomes prepared? Why egg phosphatidylcholine is used, why not other phospholipids? Egg phosphatidylcholine is more expensive than other sources of phospholipids, particularly for the final formulation such as tablets. Please explain the negative surface charge of the vesicles formed.

There are also other studies performed with liposomes prepared in the dry/solid form and compressed in tablets. I could not find that the authors have mentioned that the tablets of phospholipid-based nanosystems have already been evaluated.

Author Response

The submitted manuscript deals with development and evaluation of tablets containing lyophilized liposomes for sublingual drug delivery. The topic is interesting and fits under the scope of the Journal. However, there are some deficiencies that should be adequately addressed to strengthen the paper.

We appreciate very much the comments and suggestions from the reviewer, and we thank the opportunity for improving the manuscript.

Line 16: Check the amount of egg phosphatidylcholine and correct.

Reply: Thanks for the correction. In fact, it was a mistake and this is 7/3. This has been corrected in the revised version. Line 16.

Some numerical data should be added in the abstract to confirm the results obtained.

Reply: values of the permeability efficiency for the liposome with and without additives (0.72 ± 0.34 µg/cm2/min and 4.18 ± 0.95 µg/cm2/min, respectively) have been included in the abstract. Lines 24-26.

Liposomes cannot be produced only by dispersing phosphatidylcholine and cholesterol in water. This part in the preparation method is not clear and need clarification. How could you be sure that the liposomes are formed? Is there any picture showing the formed liposomes? Moreover, provide more data about the content of phosphatidylcholine in egg phosphatidylcholine obtained from Sigma Aldrich.

Reply: In fact, liposomes cannot be produced only by dispersing phosphatidylcholine and cholesterol in water. In material and methods section, it was explained that the lipid dispersion underwent ultrasonic agitation for 20 min at 40ºC and then was extruded through 0.45 µm. Under ultrasonic agitation at 40ºC (above transition temperature) EPC molecules are in the liquid state with mobility for bilayer arrangement and vesicle formation. Our research group has wide expertise in producing liposomes by this method. First, we used the thin-film method for liposome production and published data are available (De Jesus Valle et al., 2013 doi: 10.1038/ja.2013.32).  By a comparative study carried out in 2015, (De Jesus Valle and Sanchez Navarro, 2015 doi:10.2174/1573412910666141114221935) we proved that EPC liposomes can be obtained by the method applied here. The limitation found was that large liposomes (Dh>400 nm) were not produced. As we are not interested in producing liposomes with Dh> 400nm, we have applied the method based in ultrasonic agitation for the last years.

In the present manuscript the method used to prepare the liposomes was shortly described since a reference of previous paper was included. Nevertheless, a paragraph has been added in the new version to clarify this point. Lines 106-111.

Figure 11 shows the formed liposomes. Image A corresponds to original liposomes and images B and C to rehydrated liposomes from swelling and liposome release samples.

Sigma Aldrich.product used contains ~70% EPC

Figure 1: to increase the clarity of presentation, please explain meaning of abbreviations used in the figure in the figure’s caption (legend).

Reply: The meaning of abbreviations has been included in the caption legend of figure 1.

The DLS and ZP measurements were done in water, why not in medium simulating sublingual application?

Reply: Samples used for DLS and ZP were in citrate buffer with or without the additives. Water was used for sample dilution. The aim was to know Dh and zeta potential of produced liposomes. On top of that, images of liposomes in salivary fluid are shown in Figure 11 (image B). We agree with the reviewer that it is very interesting to know the influence of changes in pH of salivary fluid on tablet behavior and we will consider the comment for the next s study.

Line 250: indicate the pH of PBS

Reply: pH of PBS was 7. This has been indicated in the revised version. Line 250.

The abbreviation for size (line 124) is Dh, later in Results and discussion (line 289) abbreviation Dm corresponds to size. Please correct. Does the term size correspond to average (mean) diameter? If so, then mean diameter instead of size should be denoted in subchapter 2.2.3.

Reply: Thank you for comment. In fact, abbreviation (Dm) in Results and discussion is an error. The hydrodynamic diameter was measured by DLS and therefore Dh should be denoted all along the manuscript. This has been corrected in the revised version. Lines 289, 291 and table 1.

Tables: indicate n value for results expressed as average ± S.D.

Reply: Thanks for the comment. Number of replicates (n) has been included in tables and figures.

Have you checked the size of liposomes formed after disintegration of tablets? Liposomes are formed in situ after disintegration of the tablets, and then, the effect of liposomes could be expected. Do you expect disintegration of tablets? The authors performed in vitro release studies.

Reply: Liposomes formed after disintegration of tablets have been checked by SEM. The morphology of liposomes in samples from tablet swelling assay and samples from liposome release assay are shown in Figure 11 (B and C images, respectively) and these were compared to original liposomes (image A), Images reveal that particles of the same morphology as original liposomes are found in these samples. Dh was not determined because in these samples there are also particles from tablet disintegration (as seen in the images) and therefore the Dh cannot be determined accurately

With respect to the expectation of tablet disintegration, the in vivo assay performed with healthy volunteers showed that tablets disintegrated in the sublingual cavity after a period of time (data in table S2 of supplementary material). Therefore, the answer is yes, the tablets are supposed to be disintegrated in the sublingual cavity after ~60 min

What is the goal of the developed system, slow or rapid release, what do you expect from the liposomes prepared? Why egg phosphatidylcholine is used, why not other phospholipids? Egg phosphatidylcholine is more expensive than other sources of phospholipids, particularly for the final formulation such as tablets. Please explain the negative surface charge of the vesicles formed.

Reply: The goal of the developed system is a release time of ~60 min. For sublingual route, this time may be considered slow release since most sublingual formulations aim to immediate release (1-2 min). The developed system is proposed as an alternative to oral route and a disintegration time of 60 min seems reasonable for this purpose.

The reasons for using EPC is low transition temperature that allows preparation of liposomes at 40 ºC, while other phospholipids oblige to work at higher temperature due to higher transition Tª.   Thinking on loading liposomes with drugs, temperature is relevant since many drugs are thermosensitive.

With respect to negative zeta potential. In fact, phosphatidylcholines (PCs) are zwitterionic products and these are considered neutral lipids. Nevertheless, extensive literature data confirm the negative surface of PC liposomes. Commercial products are not pure PC, but a mixture of lipids with predominance of PC. The Sigma product used here had  ~70% of PC, and this may be the reason for potential zeta highly negative . A paragraph has been included in the revised version to clarify this point. Lines 293-297.

There are also other studies performed with liposomes prepared in the dry/solid form and compressed in tablets. I could not find that the authors have mentioned that the tablets of phospholipid-based nanosystems have already been evaluated.

Reply: As far as we know there are not literature information about lyophilized liposomes into tablets, except for a previous paper from our group that described vaginal tablets containing lyophilized albusomes (albumin-liposomes-microparticles).

Reviewer 2 Report

The authors present an interesting manuscript focussed on the development of a mucoadhesive vehicle based on liposomes for drug delivery at the sublingual mucosa.

It is an original approach and the work contains an appropriate volume of experiments that support the conclusions. However, there are several points to be addressed before the acceptance of the manuscript.

1. The idea of freeze-dried liposomes to obtain a long-term shelf life formulations has been developed almost since its discovery more than 60 years ago. The novelty of the present research is to include the lyophilized liposomes into another vehicle. The ideas is good considering the ability of liposomes for controlled drug delivery and the ability to encapsulate both, hydrophilic and lipophilic drugs. However in the present manuscript, the reconstitution of the liposomes is expected to be reached once administered. Isn’t it? And is it so?

2. Line 16: There is a mistake in the molar ratio Pc/Chol it should be 7/3 instead of 0.7/3.

3. Line 101: why to sonicate at 40 oC if Tm of the mixture is much lower?

4. Line 124: Dh refers to Hydrodynamic diameter but it should be defined the first time for readers not familiar with the measurement.

5. Point 2.2.4. Line 137. A capillary viscosimeter is aimed for Newtonian fluids. Why to use it for liposomes? Do they behave as ideal Newtonian?

6. Line 290. The authors find zeta potential values highly negatives for PC/chol mixtures. How do they can explain this fact when PC is a zwitterionic phospholipid and chol doesn’t confer charge?

Author Response

The authors present an interesting manuscript focussed on the development of a mucoadhesive vehicle based on liposomes for drug delivery at the sublingual mucosa.

It is an original approach and the work contains an appropriate volume of experiments that support the conclusions. However, there are several points to be addressed before the acceptance of the manuscript.

We appreciate the reviewer comments and suggestions very much and we thank the opportunity for improving the manuscript

  1. The idea of freeze-dried liposomes to obtain a long-term shelf life formulations has been developed almost since its discovery more than 60 years ago. The novelty of the present research is to include the lyophilized liposomes into another vehicle. The ideas is good considering the ability of liposomes for controlled drug delivery and the ability to encapsulate both, hydrophilic and lipophilic drugs. However in the present manuscript, the reconstitution of the liposomes is expected to be reached once administered. Isn’t it? And is it so?

Reply: Yes, indeed the liposomes are expected to reconstitute once administered, as tablet swells, and water diffuses to the the lyophilized liposomes. Swelling and liposome release assays proved that this happens since rehydrated liposomes were found in the samples form those assays (Figure 11, images B and C).

  1. Line 16: There is a mistake in the molar ratio Pc/Chol it should be 7/3 instead of 0.7/3.

Reply: Thanks for the correction. In fact, it was a mistake, and this is 7/3. This is now correct in the revised version line 16.

  1. Line 101: why to sonicate at 40 oC if Tmof the mixture is much lower?

Reply:  We apologize, we do not understand well this question. In our study we denote Tm the transition temperature for lyophilized samples. Other temperatures studied by DSC were the Tg and T´g. but these were used for optimizing the lyophilization cycle. Liposomes were prepared in absence of additives.  Do you mind that 23-24ºC is the transition Tª for EPC and therefore lower than 40 ºC? This is right, but liposomes must be prepared at temperature above the transition of lipids. We have carried out studies at different temperature and the results (unpublished data) revealed that at temperature lower than 40ºC extrusion was difficult and PDI was higher. Therefore, we selected 40ºC as the optima temperature. A paragraph explained this point has been included in lines 106-111.

  1. Line 124: Dh refers to Hydrodynamic diameter, but it should be defined the first time for readers not familiar with the measurement.

Reply: Thank you for the comment. Dh has been defined in the revised version, line 130.

  1. Point 2.2.4. Line 137. A capillary viscosimeter is aimed for Newtonian fluids. Why to use it for liposomes? Do they behave as ideal Newtonian?

Reply: We appreciate very much this comment. Low viscosity was expected for liposome suspension without additives and we assumed Newtonian behavior, but this is not correct. We tried to determine the viscosity of liposome suspension without additives using the rotary viscosimeter, but our model does not allow accurate results for samples of viscosity lower than 3 cP. Therefore, in the revised version the value of 1.17±0.03 cP has been removed and substituted by <3 cP. Corrections have been made in Material and a Methods (lines 141-147) and results (Table 1 and line 467).

  1. Line 290. The authors find zeta potential values highly negatives for PC/chol mixtures. How do they can explain this fact when PC is a zwitterionic phospholipid and chol doesn’t confer charge?

Reply:  In fact, phosphatidylcholines (PCs) are zwitterionic products and these are considered neutral lipids. Nevertheless, extensive literature data confirm the negative surface of PC liposomes prepared from commercial products. Commercial products are not pure PC, but a mixture of lipids with predominance of PC. the Sigma product used here had ~70% of PC, and this may be the reason for zeta potential highly negative. A paragraph has been included in the revised version to clarify this point. Lines 293-297.

Reviewer 3 Report

Being a pre-formulative study, the manuscript is well structured and thought out, with a significant scientific impact. 
However, several clarifications and clarifications should be provided.

line 16 - In the abstract section is described a liposomal formulation egg phosphatidylcholine and cholesterol  in 0.7/3 molar ratio, data not listed in other sections and conceptually uncorrected.

Liposomes can be realized with different phospholipids and with different amount of cholesterol. A rationale of choosing a 7:3 molar ratio should be provided, also mentioning study investigating this parameter that allow avoiding to test other formulations.
(Mare et al. - Post-insertion parameters of PEG-derivatives in phosphocholine-liposomes. - Int J Pharm. 2018 Dec 1;552(1-2):414-421. doi: 10.1016/j.ijpharm.2018.10.028. Epub 2018 Oct 10.)

Mucoadhesive properties should be tested in vitro by using suitable technique for nanoparticles and described in litterature, such mucine assay.

(Iannone M. et al. - Characterization and in vitro anticancer properties of chitosan-microencapsulated flavan-3-ols-rich grape seed extracts - Int J Biol Macromol. 2017 Nov;104(Pt A):1039-1045.  doi:10.1016/j.ijbiomac.2017.07.022. Epub 2017 Jul 4.)

Considering the egg phosphatidylcholine zwitterionic structure and the dilution medium used for DLS analyses (MilliQ water), Z-potential values should not be always extremely negative. In addition to this, lactose, mannitol, carboxymethylcellulose and alginate, never induce significant variations on this important parameter. This data should be checked again or a rationale to this strange phenomenon should be provided with suitable references.

According to this reviewer a more detailed discussion should be provided for results obtained. For example, egg phosphatidylcholine have a really low phase transition temperature. The addition of cholesterol proved to change this parameter, but it is not possible to overturn this parameter, even in formulation with high amount of colesterol (7:3 molar ratio).

The effects of pH and proteins on the liposomal formulation should be considered and discussed, especially for lyophilized liposomes. In fact, saliva is known to have:

- an acid pH which in long-term exposure damages the liposomal membranes

- high protein content, which through the phenomenon of corona proteins, could change the pharmacokinetics and pharmacodynamics of the bioactives contained in the liposomes.

All these parameters should be evaluated with proper reference.

Author Response

Being a pre-formulative study, the manuscript is well structured and thought out, with a significant scientific impact.

However, several clarifications and clarifications should be provided.

We appreciate the reviewer comments and suggestions very much and we thank the opportunity for improving the manuscript

line 16 In the abstract section is described a liposomal formulation egg phosphatidylcholine and cholesterol  in 0.7/3 molar ratio, data not listed in other sections and conceptually uncorrected.

Reply: Thanks for the correction. In fact, it was a mistake, and this is 7/3. This has been corrected in the revised version, line 16.

Liposomes can be realized with different phospholipids and with different amount of cholesterol. A rationale of choosing a 7:3 molar ratio should be provided, also mentioning study investigating this parameter that allow avoiding to test other formulations.

(Mare et al. - Post-insertion parameters of PEG-derivatives in phosphocholine-liposomes. - Int J Pharm. 2018 Dec 1;552(1-2):414-421. doi: 10.1016/j.ijpharm.2018.10.028. Epub 2018 Oct 10.)

Reply: previous studies carried out by our group (unpublished data) proved that a 7/3 ratio for EPC/Chol is the most beneficial in terms of stability. Since this finding is in accordance with comments in paper from Mare R.  et al, (International Journal of Pharmaceutics, 2018) a paragraph and the reference have been included in the revised version to justify the 7/3 ratio. Lines 108-111 and reference [30].

Mucoadhesive properties should be tested in vitro by using suitable technique for nanoparticles and described in literature, such mucine assay. (Iannone M. et al. - Characterization and in vitro anticancer properties of chitosan-microencapsulated flavan-3-ols-rich grape seed extracts - Int J Biol Macromol. 2017 Nov;104(Pt A):1039-1045.  doi:10.1016/j.ijbiomac.2017.07.022. Epub 2017 Jul 4.)

Reply: Mucoadhesion was tested in healthy volunteers (section 2.2.9.3 : Tablet test in healthy volunteers, in material and methods; results in Table S2). Therefore, we think that in vitro assay is not necessary.

Considering the egg phosphatidylcholine zwitterionic structure and the dilution medium used for DLS analyses (MilliQ water), Z-potential values should not be always extremely negative. In addition to this, lactose, mannitol, carboxymethylcellulose and alginate, never induce significant variations on this important parameter. This data should be checked again or a rationale to this strange phenomenon should be provided with suitable references.

Reply In fact, phosphatidylcholines (PCs) are zwitterionic products and these are considered neutral lipids. Nevertheless, extensive literature data confirm the negative surface of PC liposomes prepared from commercial products. Commercial products are not pure PC, but a mixture of lipids with predominance of PC. the Sigma product used here had ~70% of PC, and this may be the reason for zeta potential highly negative. A paragraph has been included in the revised version to clarify this point. Lines 293-297.

According to this reviewer a more detailed discussion should be provided for results obtained. For example, egg phosphatidylcholine have a really low phase transition temperature. The addition of cholesterol proved to change this parameter, but it is not possible to overturn this parameter, even in formulation with high amount of colesterol (7:3 molar ratio).

Reply: We agree that phase transition temperature of EPC is low, and this might be a handicap for long circulating liposomes. Nevertheless, the vehicle developed here is proposed for rapid liposome tissue uptake. EPC liposomes with 30% molar cholesterol are stable for longer periods than expected for the tablet in the sublingual cavity. Therefore, we think that these are suitable for this purpose.

The effects of pH and proteins on the liposomal formulation should be considered and discussed, especially for lyophilized liposomes. In fact, saliva is known to have:

- an acid pH which in long-term exposure damages the liposomal membranes

- high protein content, which through the phenomenon of corona proteins, could change the pharmacokinetics and pharmacodynamics of the bioactives contained in the liposomes.

All these parameters should be evaluated with proper reference.

Reply: A simulated salivary fluid proved to be biorelevant with respect to pH, composition, viscosity, etc. (Joseph Ali et al., 2021) was used for the swelling assay and the results revealed that liposomes hydration occurs under conditions that mimic the salivary fluid. The hypothesis was: water diffuses from salivary fluid to the tablet and lyophilized liposomes rehydrate. This was confirmed by the presence of   liposomes in the swelling assay samples (Figure 11. Image B).

We agree that changes in salivary fluid may influence the tablet swelling process and that the suggested assays are very interesting. We appreciate the comment that will be considered for further studies.

Round 2

Reviewer 2 Report

The authors have addressed the questions required and the manuscript can be accepted in the present form.

Reviewer 3 Report

Thanks for the corrections made and for the clarifications provided to this reviewer.
Congratulations on the manuscript of excellent scientific impact.